# Effect of Bainitic Isothermal Treatment on the Microstructure and Mechanical Properties of a CMnSiAl TRIP Steel

Alexis Guzmán [1] and Alberto Monsalve [2,3,*]

1. Departamento de Ingeniería en Metalurgia, Facultad de Ingeniería, Universidad de Atacama, Copiapo 1531772, Chile; alexis.guzman@uda.cl
2. Departamento Ingeniería Metalúrgica, Facultad de Ingeniería, Universidad de Santiago de Chile, Santiago 9170022, Chile
3. Structural Integrity Program, Facultad de Ingeniería, Universidad de Santiago de Chile, Santiago 9170022, Chile
* Correspondence: alberto.monsalve@usach.cl

**Abstract:** TRIP-assisted CMnSiAl steels with a fully martensitic initial microstructure have been studied in order to investigate the effect of partial substitution of Si by Al. The steel was fabricated by casting in a sand mold, hot forged, homogenizing, hot rolling, cold rolling, intercritical annealing, and finally, an isothermal bainitic treatment. During the intercritical annealing at 1023 K (750 °C) for 420 s, a matrix with a microstructure consisting of 50% proeutectoid ferrite was induced and after isothermal treatment at 663 K (390 °C) for 300 s, a microstructure of ferrite, bainite, and retained austenite was obtained. This austenite was carbon enriched and therefore stabilized, with an $M_s$ lower than original austenite, before bainitic treatment. Microstructure and tensile properties have been analyzed, comparing these results with those corresponding to a commercial TRIP 780 steel. During tensile tests, retained austenite changes from 8.1 to 1.9%vol, with a total elongation of 23.2%, which demonstrates the TRIP effect. Texture analysis showed a weak γ-fiber in the steel, which deteriorates the drawing properties of the steel. The maximum elongation during the tensile test of this steel was obtained at 323 K (50 °C), correlating with the TRIP effect present in steel, which is favored when working within the critical zone of temperatures $M_s^\sigma$ and $M_d$. The results show that partial substitution of Si by Al decreases yield stress, ultimate tensile strength, increases the total elongation, and decreases the temperature for maximum elongation, related to TRIP 780.

**Keywords:** TRIP steels; retained austenite; martensite; bainite

## 1. Introduction

TRIP (Transformation Induced Plasticity) steel belongs to Advanced High-Strength Steels (AHSS) [1,2], and they are characterized by high values of strength/elongation and high strain hardening index [3], with tensile strengths up to 1200 MPa and total elongations up to 40% [2]. The microstructure consists of ferrite, bainite, and retained austenite. In some cases, martensite and carbides could be present [4]. Retained austenite can transform to martensite during deformation. This austenite to martensite transformation, delaying the necking in the tensile test, explained a high value of homogeneous deformation and, therefore, a high value of total fracture elongation [5]. Typically, the chemical composition of these kinds of steels is: 0.1 to 0.4 wt% C; 1.0 to 2.0 wt% Mn; 1.0 to 2.0 wt% Si and other elements [6]. Usually, TRIP steels are fabricated by means of hot and cold rolling followed by (a) intercritical annealing (IA) plus isothermal bainitic treatment (IBT) or (b) austenitization plus isothermal bainitic transformation. Recently, Chen et al. [7] have studied the fabrication of TRIP steels through a homogenizing, hot rolling, fast cooling (air or water cooling) following maintenance at 400 °C for one hour in order to simulate the coiling stage in an industrial fabrication process, obtaining TRIP steels with UTS higher than 1150 MPa. About the influence of alloying elements, carbon increases the stability of

the retained austenite, manganese increases the solid solution strengthening, and silicon suppresses the formation of cementite during the IBT [6]. Similar to silicon, aluminum stabilizes ferrite, although Si has the disadvantage of forming oxides such as $FeSiO_4$ and $SiO_2$ that decrease the surface quality of the steels [8]. Aluminum also decreases the density of steel due to Al atoms having higher a diameter than pure iron and also due to its low density itself. A key factor in TRIP aided steels is the austenite stabilization and, therefore, the prevention of athermal martensite formation during final cooling [2].

Silicon avoids the precipitation of carbides, stabilizing the retained austenite and so, attenuating the TRIP effect. Furthermore, the hardening effect of Al is lower than Si, so a total replacement of Si could decrease the mechanical resistance of the steel. Therefore, only a partial substitution of Si by Al is possible. Shi et al. [9] studied the effects of intercritical annealing temperature on the mechanical properties in steels with 0.5 wt% Si and 1.01 wt% Al at different annealing temperatures and 400 °C of bainitic temperature, finding yield strength and ultimate tensile strength lower than those obtained in this work. Also, Emadoddin et al. [10] have analyzed two TRIP steels with different contents of Al and Si, focusing their study on the changes in texture components due to different cold rolling conditions and annealing temperatures. Moreover, the effect of partial substitution of Si by Al was studied by De Meyer et al. [11], finding between other results, that the rate of formation of austenite during isothermal bainitic treatments is similar for both steels. A study of the influence of heat-treating conditions on the retained austenite was carried out by Jacques et al. [12], analyzing four TRIP-assisted steels with different contents of Al and Si, finding that mixed Al-Si TRIP steel shows the best combination for mechanical properties and industrial restrictions.

The aim of this work was to study a new composition of TRIP-assisted steel based on the partial substitution of Si by Al and to investigate the influence of the isothermal bainitic treatment on the microstructure and mechanical properties of the steel by comparing the results with the properties of a commercial TRIP 780 steel.

## 2. Materials and Methods

The steel used in this research was fabricated in the Foundry Laboratory of the Metallurgical Engineering Department of the Universidad de Santiago de Chile (Santiago, Chile). An induction furnace was used, and the steel was poured into a sand mold in order to obtain an ingot of 25 Kg and a size of 100 mm × 100 mm × 320 mm. The raw materials were wire for the electrode, ferromanganese (70–75 wt% Mn), ferrosilicon (70–75 wt% Si), and pure Al. Table 1 shows a comparison between the studied steel and TRIP 780 commercial steel. The chemical analysis was made in a spark emission spectrometer Spectro, model SPECTROMAXx (Spectro Analytical Instruments GmbH, Kleve, Germany) in accordance with ASTM E415 standards [13].

**Table 1.** Chemical compositions of the studied steel and TRIP 780 commercial steel (in wt%).

| Steel | C | Si | Mn | Al | $p$ |
|---|---|---|---|---|---|
| Studied steel | 0.21 | 0.82 | 1.87 | 0.43 | 0.03 |
| TRIP 780 | 0.20 | 1.53 | 1.92 | 0.00 | 0.01 |

The ingot was hot forged at 1373 K (1100 °C) by means of a hydraulic hammer, obtaining a billet of 18 mm × 140 mm × 1000 mm. Then, the billet was mechanized, obtaining samples of 17 mm × 51 mm × 138 mm, which were homogenized at 1423 K (1150 °C) for 30 min, followed by hot rolling in multiple steps until achieving a 5 mm thickness, corresponding to a reduction of 70%. The thermal registration was carried out by means of an optical pyrometer and a type K thermocouple welded to the samples, taking care that the temperature was higher than 1123 K (850 °C). Then, after hot rolling was complete, the samples were cooled inside the furnace in order to obtain a ferritic-pearlitic microstructure.

After finishing the hot rolling process, the samples were pickled using hydrochloric acid (HCl) in order to remove the surface oxide. Then, samples were cold rolled in two steps: the first reduced the thickness to 1.7 mm (68% reduction) followed by stress-relieving annealing at 873 K (600 °C) for 1 h followed by air cooling, and the second reduced the thickness to 0.5 mm (70% reduction).

The heat treatment to induce the TRIP behavior is shown in Figure 1. The first stage consisted of obtaining a fully martensitic initial microstructure, for which the samples were heated to 1223 K (950 °C) for 15 min, followed by quenching in water at room temperature. Then, an intercritical heat treatment was carried out in order to obtain a 50% proeutectoid ferrite microstructure. To determine the conditions of this treatment, a kinetic study of the microstructure evolution was carried out using samples of 20 mm × 20 mm × 0.5 mm, using different times at a temperature of 750 °C (0, 20, 40, 60, 120, 300, 420, and 1000 s). From these heat treatments, it was concluded that a time of annealing of 420 s at 1023 K (750 °C), produced a 50% ferrite microstructure after cooling at room temperature.

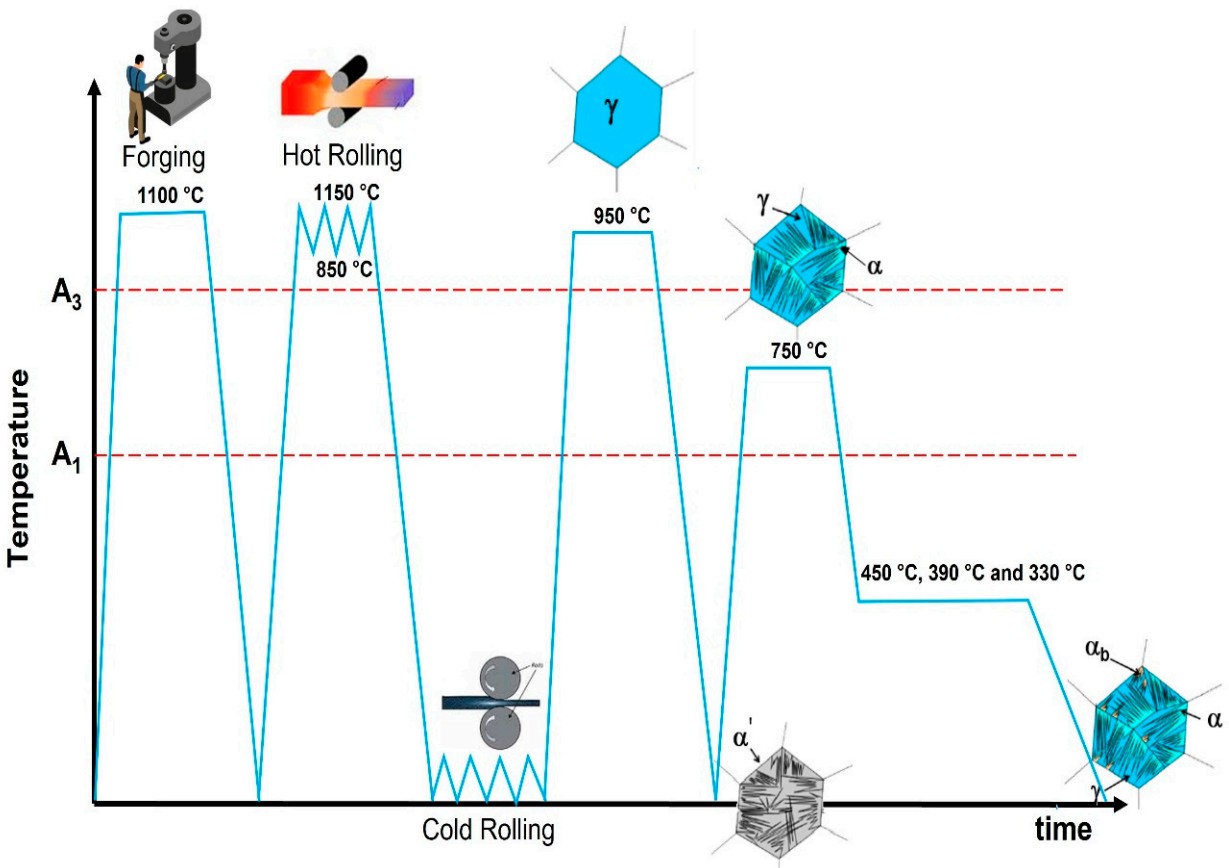

**Figure 1.** Thermomechanical treatment used in order to induce the TRIP microstructure.

A second kinetic study was carried out during isothermal bainitic treatment in order to analyze the retained austenite and the carbon enrichment. To do this, three temperatures were defined: 603 K, 663 K, and 723 K (330 °C, 390 °C, and 450 °C), during different times: 60, 120, 180, 300, and 600 s. The samples were analyzed using optical and electronic microscopy and X-ray diffraction.

The critical temperatures of the steel ($A_1$ and $A_3$) were determined by differential scanning calorimetry (DSC) using samples of 0.1 g, heating them at a rate of 20 °C/s up to 1273 K (1000 °C) in a nitrogen-rich atmosphere (TA Instruments, model 2960 Dual DSC-TG, Newcastle, DE, USA). Ms and Bs were determined using empirical equations, which depend on the chemical composition of the steel.

Metallographic analysis was carried out in accordance with the ASTM E3 standard [14], consisting of grinding using SiC paper followed by alumina (1 and 0.5 μm) polishing. The etching was made using the ASTM E407 [15] by means of the following etching solutions: nital 3% (nitric acid 3% *v/v* in ethyl alcohol), picral 4% (picric acid 4% *v/v* in ethyl alcohol), sodium metabisulfite 10% *v/v* in distilled water), and LePera (picral 4% *v/v* plus sodium metabisulfite 1% in distilled water mixed in equal parts). Optical microscopy was carried out in an Olympus microscope model BX51 (Olympus, Tokyo, Japan) with polarized light, a scanning electron microscope Leo model 1420 VP (LEO Electron Microscopy Inc., New York, NY, USA), an atomic force microscope Veeco Nano Scope model 3A (Veeco, New York, NY, USA) and a diffractometer Siemens Kristalloflex D5000 (Brücker Belgium SA/NV, Kontich, Belgium), with the following characteristics: Cu Kα radiation with λ = 1.5406 Å, 30 mA, 40 kV, step 0.02° and counting time per step: 10 s. Texture analyses were made by using the scanning electronic microscope FEI Nova 600 Nanolab Dual Beam FIB-SEM using the electron backscattering diffraction technique (FEI Company, Hillsboro, OR, USA). The following are the set of parameters used in this analysis: a voltage of 20 kV, an inclination angle of 70°, and a step between 0.02 to 0.2 μm over a square grid. For this analysis, samples were mechanically polished, then electropolished, and a final polishing stage using silica colloid (0.035 μm) for 10 min.

Mechanical characterization was carried out through a tensile test using a Tinius-Olsen servohydraulic tensile machine model Super L (Tinius Olsen, Horsham, PA, USA); according to the ASTM E8 standard [16], the anisotropy tests were made according to ASTM E517 standard [17].

## 3. Results

In order to define the heat treatment conditions, the critical temperatures for phase transformations $A_1$ and $A_3$ were determined by means of differential scanning calorimetry (DSC) and thermal analysis of the cooling curves of samples heated to 1423 K (1150 °C). These temperatures were compared with those computed through empirical equations based on the main alloying elements of the steel. Five samples were studied using the DSC technique, determining that temperatures $A_1$ and $A_3$ were 999 K and 1113 K (726 °C and 840 °C), respectively. For thermal analysis, samples were austenitized at 1423 K (1150 °C) and cooled inside the furnace. From cooling curves analysis, temperatures $A_1$ and $A_3$ were determined as 1003 K and 1118 K (730 °C and 845 °C), respectively. In Table 2, temperatures obtained by means of DSC, thermal analysis, and empirical equations were compared. Furthermore, in Table 3, the bainitic (Bs) and martensitic (Ms) start temperatures obtained by empirical equations are shown [18].

**Table 2.** Values of $A_1$ and $A_3$, experimentally obtained and computed by means of empirical equations [18].

| Technique or Reference | Temperature (°C) | |
|---|---|---|
| | $A_1$ | $A_3$ |
| DSC | 726 | 840 |
| Thermal Analysis | 730 | 845 |
| Andrews | 727 | 857 |
| Grange | 714 | 835 |
| Kasatkin et al. | 728 | 870 |

**Table 3.** Bs and Ms temperatures computed by means of empirical equations [18].

| Technique or Reference | Temperatura (°C) | |
|---|---|---|
| | Bs | Ms |
| Andrews | - | 268 |
| Lee et al. | 515 | - |
| Kirkaldy et al. | 631 | - |
| Payson and Savage | - | 347 |
| Steven and Haynes | 622 | 256 |

Once critical temperatures were defined, the goal was to induce a martensitic microstructure. To do this, a heat treatment consisting in heating the samples at 1223 K (950 °C) for 15 min, followed by a water cooling, was carried out. Figure 2 shows the microstructure attained, consisting of lath martensite.

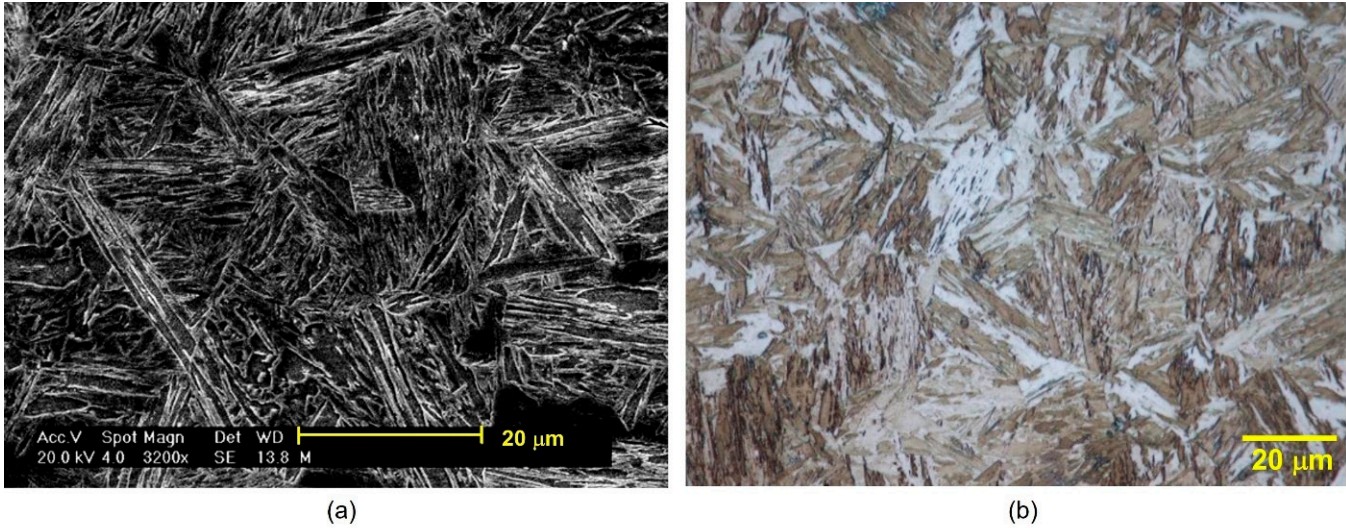

(a)                                                        (b)

**Figure 2.** Initial martensitic microstructure: (**a**) Scanning electronic microscopy using secondary electrons; (**b**) Optical image of the microstructure.

After the martensite microstructure was induced, an intercritical annealing heat treatment at 1023 K (750 °C) for 420 s was carried out, obtaining a biphasic microstructure containing 50% of polygonal ferrite plus 50% austenite (which transforms to martensite during cooling), as can be observed in Figure 3.

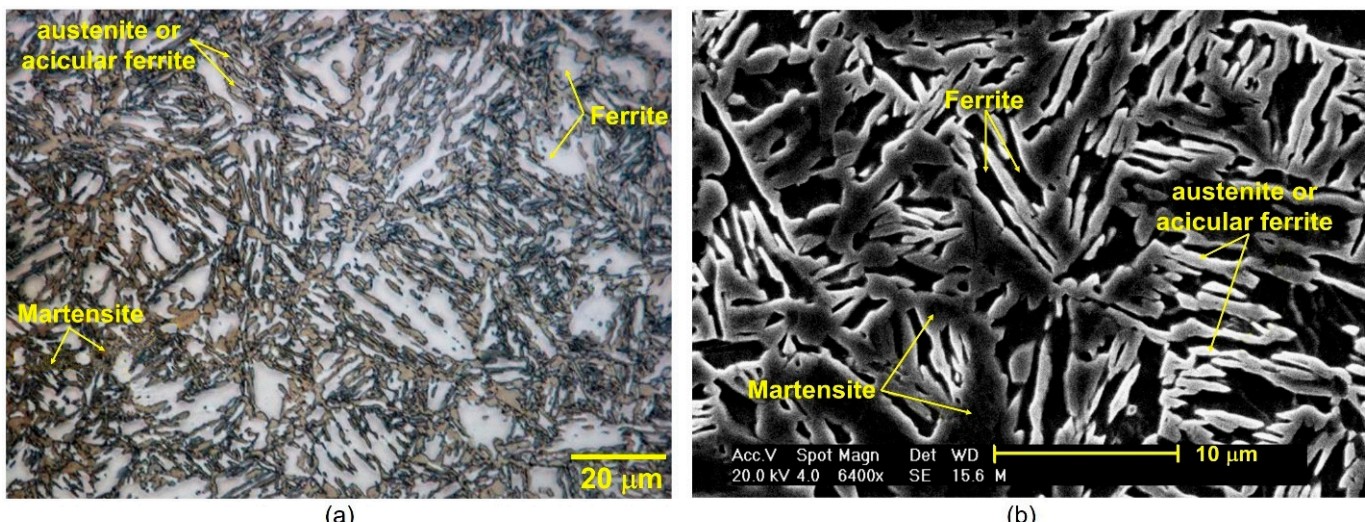

(a)                                                        (b)

**Figure 3.** Microstructure obtained during intercritical annealing treatment at 1023 K (750 °C) for 420 s, etched with nital 3% (**a**) Optical microscopy and (**b**) Electron microscopy.

X-ray diffraction was used for the determination of retained austenite obtained from different bainitic isothermal treatments. The quantification was carried out using a direct comparison method, which is based on the proportionality between the intensities of a phase in a mixture of phases and the volume concentration (Steven y Haynes del 56). For this, the equation used by De Meyer [19] and the integrated intensities of peaks

corresponding to (110), (200), (211), and (220) of ferrite and (111), (200), (220), and (311) of austenite were utilized, in accordance to:

$$V_\gamma = \frac{1.4I_\gamma}{I_\alpha + 1.4I_\gamma}$$

where:

$I_\gamma$ is the average intensity of austenite peaks
$I_\alpha$ is the average intensity of ferrite peaks.

Carbon dissolved in austenite was computed by means of:

$$\%C = \frac{a_\gamma - 0.35550}{3.8 \cdot 10^{-3}}$$

With $a_\gamma$ the lattice parameter of austenite in nm.

The austenite fraction measured and the carbon enrichment are shown in Figures 4 and 5, respectively, where it can be observed that the best combination between both parameters corresponds to a heat treatment at 663 K (390 °C) for 300 s (M390-300). The retained austenite volume was 8.1% with a 1.3% C. A greater time of heat treatment does not significantly change the austenite fraction and its carbon content.

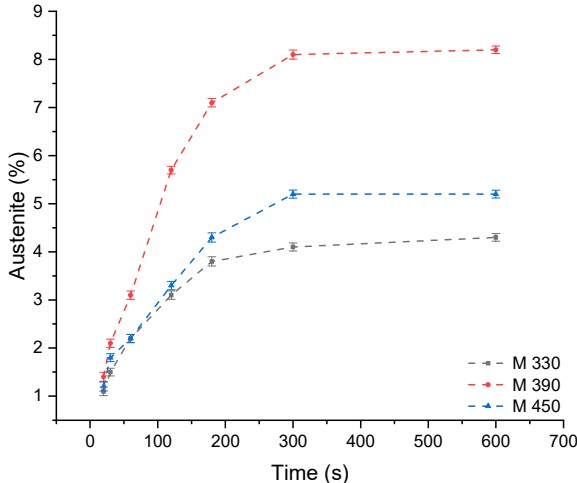

**Figure 4.** Variation of austenite volume against temperature and soaking time during bainitic isothermal treatments: M 603 K (330 °C), 663 K (390 °C), and 723 K (450 °C).

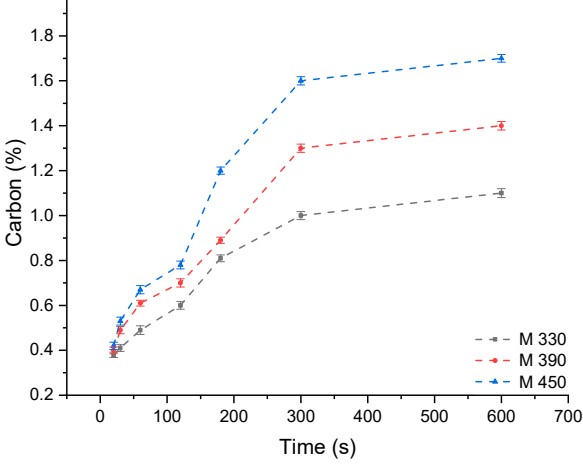

**Figure 5.** Variation of the carbon enrichment of austenite against temperature and soaking time during bainitic isothermal treatments: 603 K (330 °C), 663 K (390 °C), and 723 K (450 °C).

Due to sample M390-300 showing the best combination between austenite volume and carbon content, a microstructural study by means of OM and SEM was carried out. The microstructure of this sample submitted to a bainitic isothermal treatment at 663 K (390 °C) for 300 s (M390-300) is shown in Figure 6. The etching was made with LePera reagent, showing a ferritic matrix (dark brown phase) plus some austenite or martensite (white phase). Figure 7 shows the same sample etched sequentially with sodium metabisulfite (10% in distilled water) followed by picric acid (4% in ethanol). It is possible to distinguish the presence of ferrite (green), bainite (brown), and austenite or martensite (white). By means of quantitative optical microscopy, the fraction of polygonal ferrite (formed during intercritical annealing) was estimated as 57%. The other phases, mainly consisting of martensite, austenite, and acicular ferrite, were determined to be 43%. The dark microconstituent was 31.6%, so the 11.4% difference corresponds to the white phase.

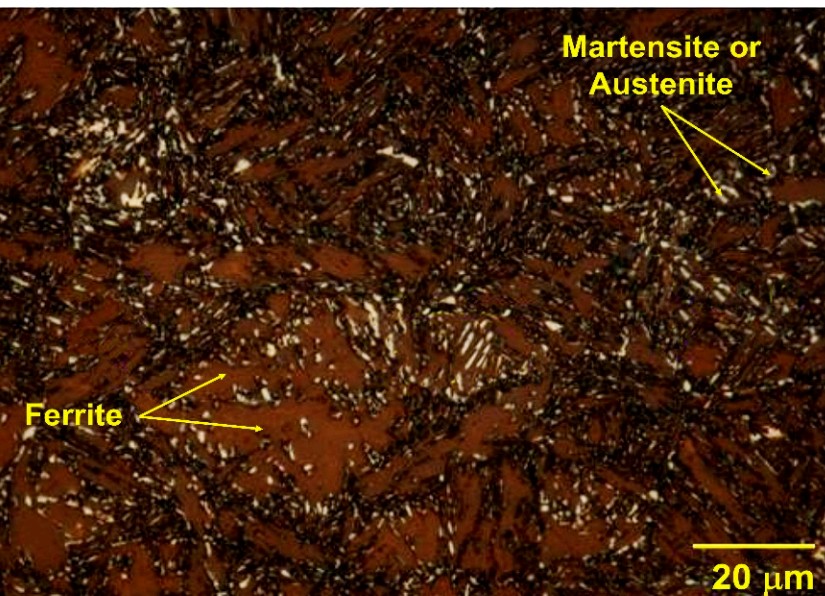

**Figure 6.** Optical micrography of M390-300 steel etched with Le Pera.

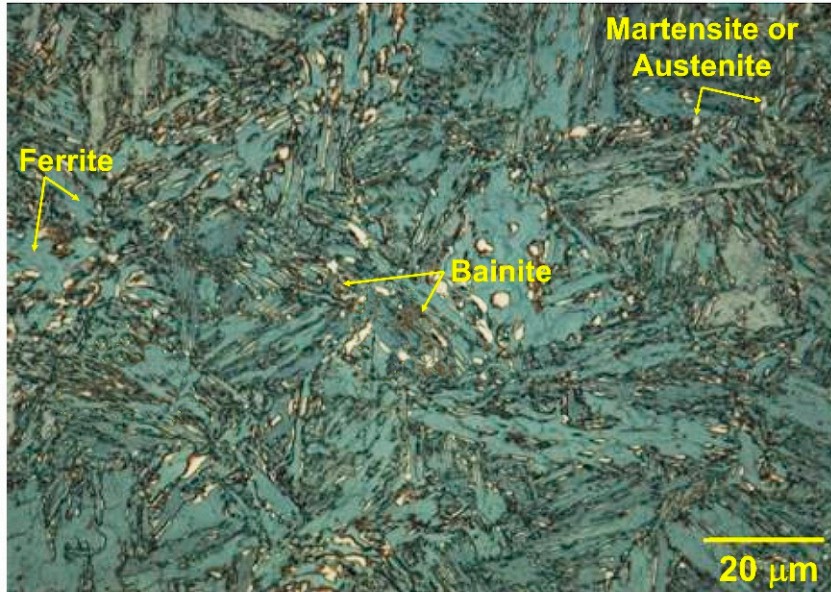

**Figure 7.** Optical micrography of M390-300 steel etched with sodium metabisulfite (10% in distilled water) plus picric acid (4% in ethanol).

Figure 8 shows a heterogeneous ferritic matrix with acicular components (ferrite or bainite) and some retained austenite, both in blocks and acicular. This kind of heterogeneous morphology is inherited from the initial martensitic microstructure, which is consistent with what has been reported by other authors [20,21].

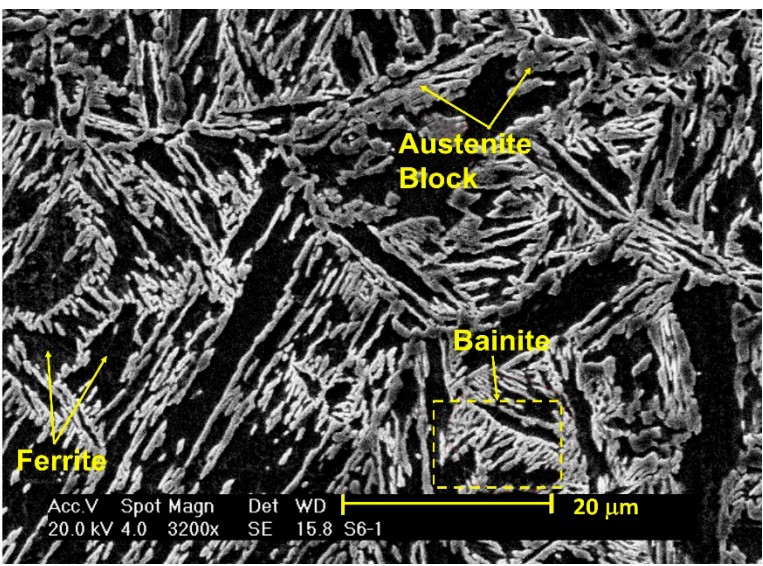

**Figure 8.** SEM image of M390-300 sample etched with nital 3%.

From the comparison of the resulting microstructures observed by OM and SEM with those obtained by the authors [8], where a similar composition but different initial microstructure were used, it can be concluded that the microstructure plays an important role in phases and its morphology. Moreover, it has a strong influence on the retained austenite and on carbon enrichment.

An EBSD study of the heat-treated samples at 723, 663, and 603 K (450 °C, 390 °C, and 330 °C) for 300 s was carried out. Figure 9 shows the OIM (Orientation Imaging Microscopy) analysis of these samples, with the inverse pole figure in order to facilitate the orientations. Acicular ferrite morphology can be observed, which is inherited from the initial martensitic (α') microstructure. Moreover, it is possible to observe that ferrite has no predominant orientation. This may be because, when the martensitic phase is induced, control over different variables that were used during the fabrication process is lost (cold rolling, the effect of cooling rate, etc.).

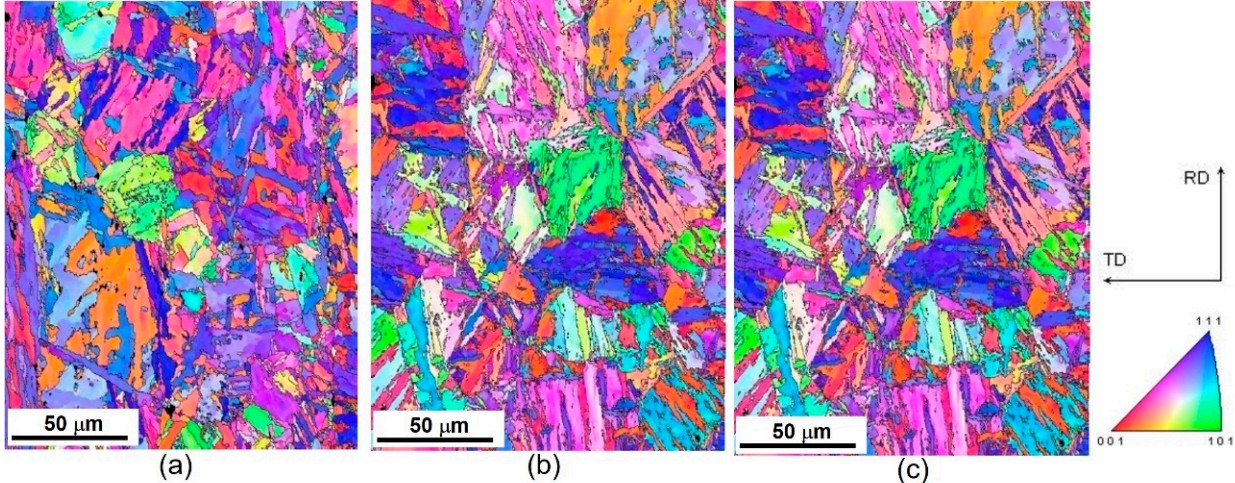

**Figure 9.** OIM analysis (**a**) M330-300 (**b**) M390-300 y (**c**) M450-300.

Figure 10 shows the orientation distribution functions (ODF) for BCC structures (mainly ferrite and bainite) for samples M330-300, M390-300, and M450-300. In these images, a weak presence of gamma fiber (<111>//ND) and alpha (<110>//ND) can be observed, which is attributed to the thermomechanical process used. It has been reported that when an AHSS steel has a weak gamma fiber, it will not have good behavior in deep drawing processes [4,22–24]. Table 4 shows a summary of the main texture components for BCC structures observed in studied steels.

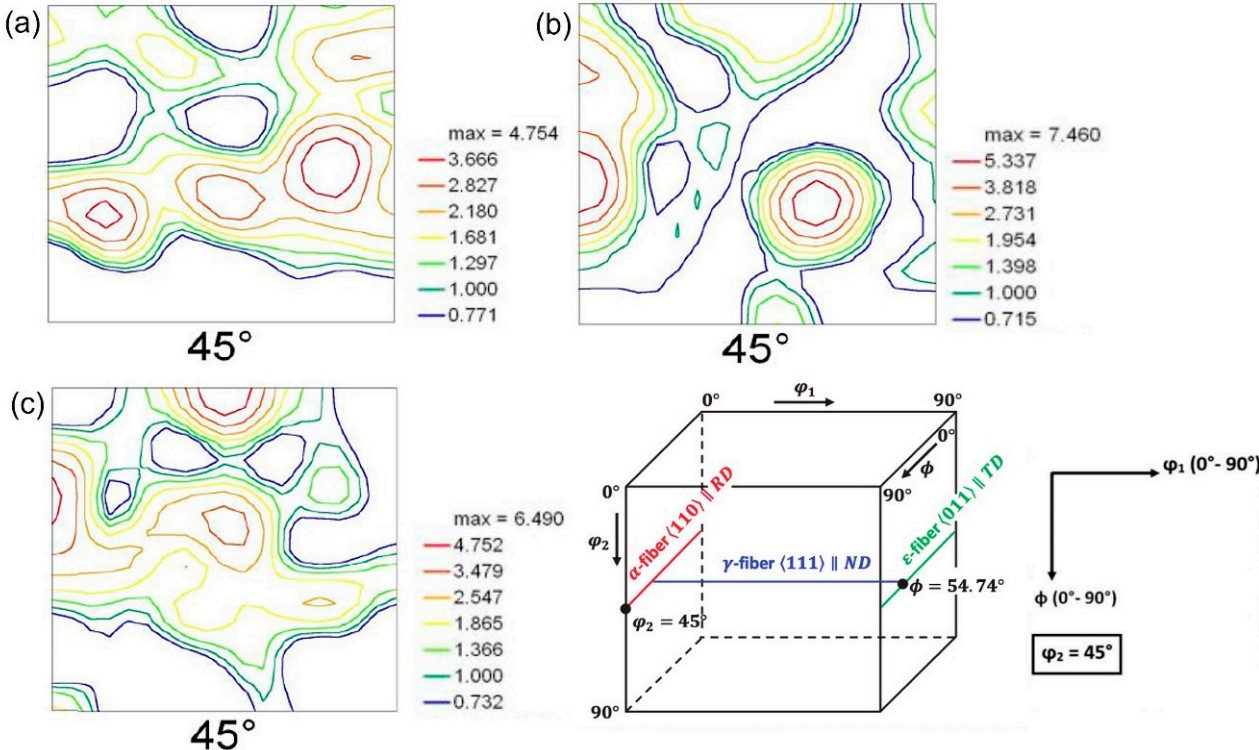

**Figure 10.** ODF's for ϕ2 = 45°: (**a**) M330-300, (**b**) M390-300 (**c**) M450-300.

**Table 4.** Main textures components showed by studied samples (ferrite and bainite).

| Samples | Texture Component | Fiber Component |
|---------|-------------------|-----------------|
| M330-300 | {111} <110> | γ-Fiber |
| | {111} <112> | γ-Fiber |
| | {001} <110> | Rotated Cube |
| M390-300 | {111} <110> | γ-Fiber |
| | {114} <110> | α-Fiber |
| | {113} <110> | α-Fiber |
| | {112} <110> | α-Fiber |
| M450-300 | {112} <110> | α-Fiber |
| | {113} <110> | α-Fiber |
| | {001} <110> | Rotated Cube |

Figure 11 shows the results of the texture analysis of retained austenite present in the microstructure. It is possible to observe that austenite appears preferentially in grain boundaries, and it has a predominant texture component <101>//ND. This kind of component was reported by Davut et al. [25] and had a beneficial effect on strain-induced martensitic transformation. Moreover, the sample with the greatest presence of austenite was M390-300, which is in accordance with the DRX analysis.

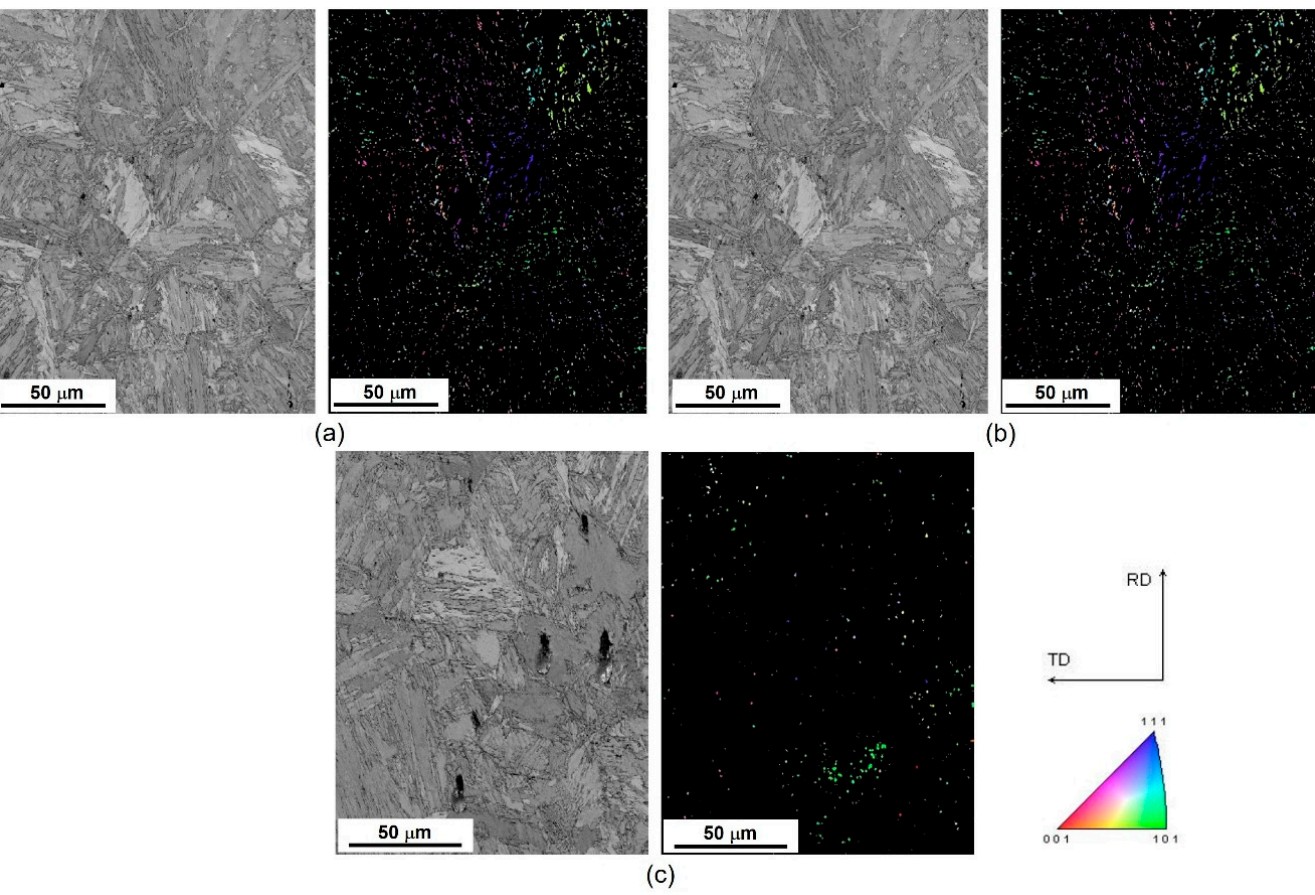

**Figure 11.** OIM analysis of austenite: (**a**) M330-300 (**b**) M390-300 y (**c**) M450-300.

Engineering stress-strain curves for samples M330-300, M390-300 and M450-300 are shown in Figure 12. It can be observed that these steels do not present the yield point phenomena, which is characteristic of low interstitial alloy steels. This behavior can be explained taken into account the high dislocation density of ferrite due to the transformation of austenite to martensite [26] and the effect of substitutional alloying elements in ferrite such as Si and Al [27].

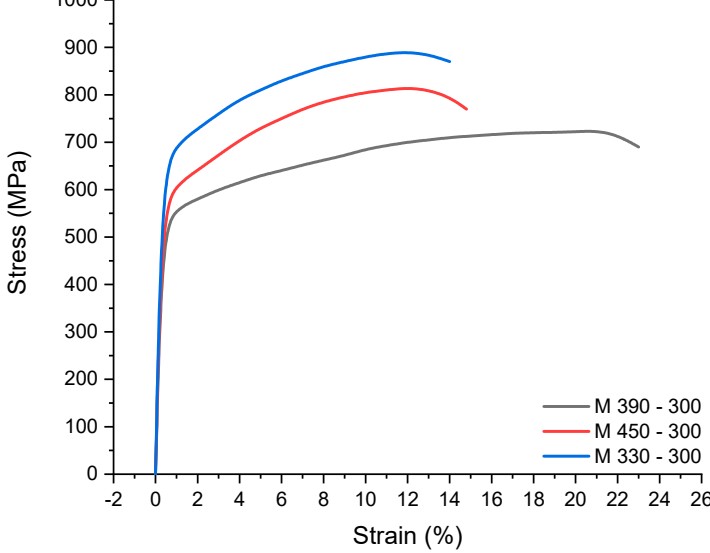

**Figure 12.** Stress-strain curve for M390-300, M450-300, and M330-300 samples.

Table 5 shows a summary of the mechanical properties of the studied steels. It is observed that the sample M390-300, despite having lower yield stress ($\sigma_0$) and ultimate tensile stress (UTS), its elongation to fracture ($\varepsilon$), and strain hardening index (n) were the highest, even comparing them with commercial TRIP 780 steel. One of the explanations for the fact that M390-300 steel presents a high elongation to fracture consists of its high content of retained austenite, which is approximately 8.1% enriched with 1.3% carbon (% by mass). This phase, when plastically deformed, will be transformed into martensite (TRIP effect,) delaying the occurrence of necking. The lowest mechanical strength values experienced by M390-300 steel compared to TRIP 780 steel can be attributed to the partial replacement of Al by Si [28]. On the other hand, r and $\Delta$r values are better in M390-300 than in TRIP 780 steel, which implies a good performance during drawing processes for the former. Moreover, the fact that the *n* value is higher in M390–300 than in TRIP 780 is related to a higher hardening capacity, which might be important in some applications, such as in the automotive industry. Finally, TRIP 780 includes a cold-rolled stage, so the comparison between both steels must consider this difference. In this work, the objective was not to enhance the TRIP 780 but only to use it as a reference.

**Table 5.** Mechanical properties of the studied steels.

| Samples | $\sigma_0$ (MPa) | UTS (MPa) | $\varepsilon$ (%) | *n* | r | $\Delta$r |
|---------|---------|-----------|-------|-----|-----|-----|
| M330-300 | 632 | 890 | 14.2 | 0.13 | 1.43 | −0.27 |
| M390-300 | 511 | 724 | 23.2 | 0.19 | 1.53 | −0.18 |
| M450-300 | 555 | 815 | 14.8 | 0.15 | 1.23 | −0.25 |
| TRIP 780 | 625 | 870 | 18.0 | 0.13 | 1.12 | −0.66 |

The results obtained through anisotropy tests are directly related to the texture components found in steels. Regarding the normal anisotropy index (r), it can be observed low values, which is explained due to the weak presence of gamma fiber (<111>//ND) and also, the texture component {001} <110> in the ferritic matrix, which is not beneficial for cold forming processes [29–31]. The negative values of the planar anisotropy index ($\Delta$r) are explained by the presence of the texture component {112} <110> in the ferritic matrix [23].

In order to corroborate the TRIP effect of the studied steels, X-ray diffraction was used. In Table 6, it can be seen that the percentage of retained austenite ($\gamma$) present in the microstructure of the steels decreases when they are subjected to plastic deformation to fracture; that is, the austenite ($\gamma$) is transformed into martensite ($\alpha'$) corroborating the presence of the TRIP effect.

**Table 6.** Retained austenite of deformed and undeformed samples.

| Samples | Austenite (% Vol.) | |
|---------|-----------|----------|
| | Undeformed | Deformed |
| M330-300 | 4.1 | 1.6 |
| M390-300 | 8.1 | 1.9 |
| M450-300 | 5.2 | 3.2 |
| TRIP 780 | 8.0 | 2.3 |

Figure 13 shows the diffractogram for sample M390-300. It is clearly seen that the peaks corresponding to the austenite phase ($\gamma$) considerably decrease their intensity when the sample is plastically deformed until fracture, giving rise to the appearance of martensite ($\alpha'$).

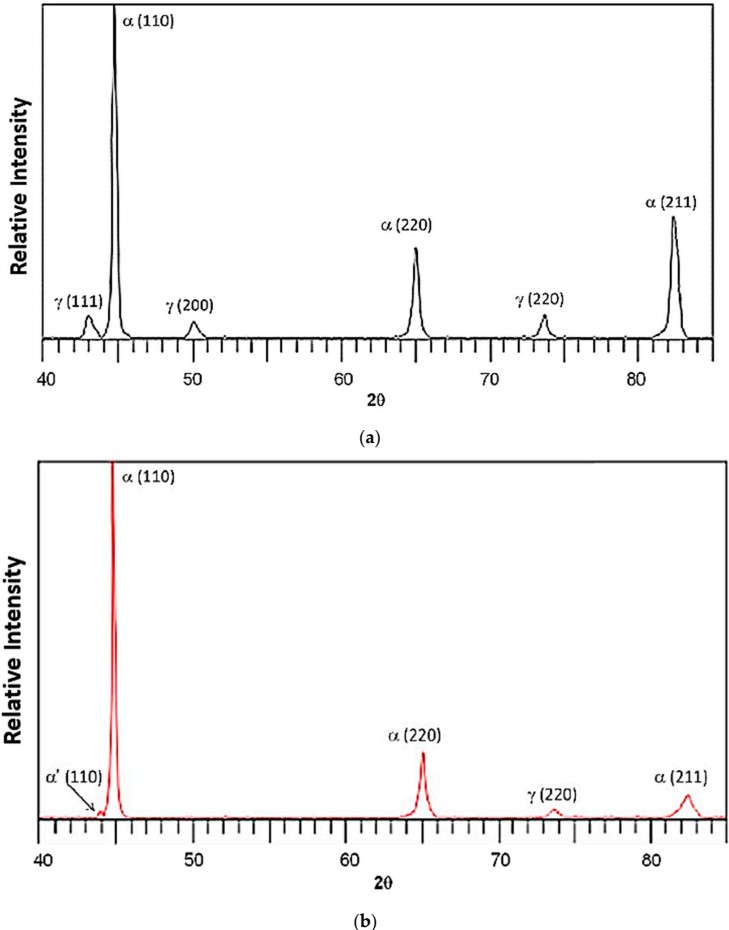

**Figure 13.** Difractograms for sample M390-300: (**a**) Without deformation y (**b**) After tensile test.

Due to the fact that the M390-300 sample showed the best combination of retained austenite and C enrichment, a thermal study of the stability of the retained austenite was carried out in order to understand its influence on the mechanical properties.

It is known that steels with TRIP behavior are very sensitive to the temperature at which they are mechanically tested because the retained austenite ($\gamma$) presents a behavior where the transformation to martensite ($\alpha'$) is strongly dependent on temperature ($M_s^\sigma, M_d$) [32]. The results of the study of the influence of temperature on mechanical properties for steel M390-300 and TRIP 780 are shown in Figure 14.

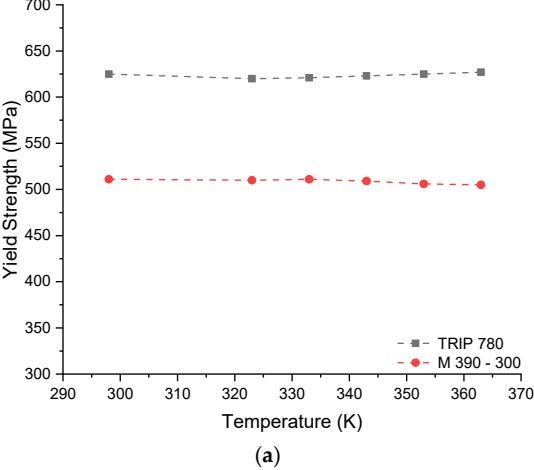

(**a**)

**Figure 14.** *Cont.*

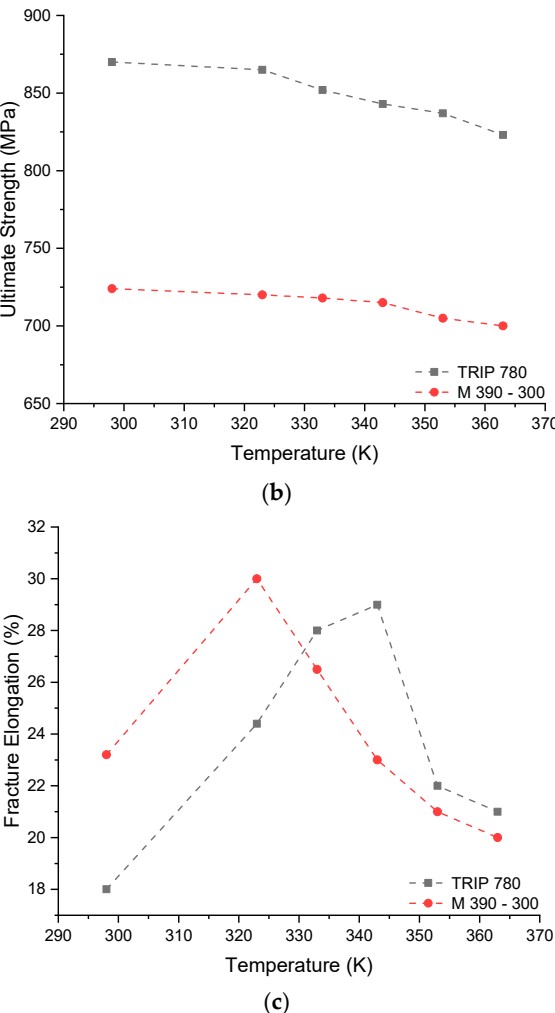

**Figure 14.** Effect of temperature on mechanical properties of M390-300 sample. (**a**) Yield stress; (**b**) UTS; (**c**) Fracture elongation.

Figure 14 shows that for both TRIP 780 and M390-300 steel, the temperature does not affect the yield stress ($\sigma_0$), but an indirect relationship between the ultimate tensile strength (UTS) and the test temperature is appreciated; that is, as the temperature increases, the maximum tensile strength decreases.

When analyzing the influence of temperature on the fracture elongation of steels, it is observed that as the temperature increases, the elongation increases to a maximum value and subsequently decreases again. The increase in elongation can be attributed to additional deformation, which is related to the Magee and Greenwood–Johnson effects [32,33], effects that have been reported by several researchers [34,35].

In addition, the critical temperatures $M_s$, $M_s^\sigma$, and $M_d$ were determined for the retained austenite of each steel, using the equations proposed by Bouquerel [36]. The results are shown in Table 7.

**Table 7.** Critical temperatures computed for retained austenite.

| Steels | Temperature (K) | | | |
|---|---|---|---|---|
| Samples | $M_s$ | $M_s^\sigma$ | $T_{max}$ Deformation | $M_d$ |
| M390-300 | 270 | 283 | 323 | 353 |
| TRIP 780 | 253 | 288 | 343 | 363 |

When analyzing Figure 14c, it can be seen that the steels M390-300 and TRIP 780 achieve their maximum elongation of 30% and 29% at 323 K (50 °C) and 343 K (70 °C), respectively. When comparing the temperature at which elongation is at a maximum with the temperature ranges obtained in Table 7 through the equations of De Cooman and Bouquerel, it is observed that this temperature is within the range that maximizes the TRIP effect ($M_s^\sigma$ and $M_d$). At higher temperatures, the elongation decreases considerably because it approaches the stability zone of the retained austenite ($\gamma$), an area in which the retained austenite ($\gamma$) is so stable that it does not transform to martensite ($\alpha'$); it only deforms ($M_d$).

**4. Conclusions**

In this work, a typical composition of steel with TRIP behavior was used, partially replacing Si with Al. In addition, an experimental procedure was used that included inducing an initial martensitic microstructure ($\alpha'$) followed by intercritical annealing and bainite isothermal treatments. The main objective of this work was to develop steel that presented the TRIP phenomenon to study microstructural features, tensile testing, texture components, and thermal stability of retained austenite. The main conclusions of this work are:

a.  Heat treatment consisting of intercritical annealing at 1023 K (750 °C) for 420 s, followed by an isothermal bainitic treatment at 663 K (390 °C) for 300 s, generated a multiphase microstructure composed of 57% of polygonal ferrite plus a mixture of 43% of martensite, acicular ferrite and retained austenite. In addition, this heat treatment presented the best balance between retained austenite fraction and carbon enrichment, which was 8.1% vol. and 1.3% mass, respectively.
b.  Related to the texture components, a weak gamma fiber (<111>//ND) was observed, which decreased with the increase in temperature during the isothermal bainitic treatments. This weak presence may be due to the fact that by inducing the initial martensitic microstructure ($\alpha'$), all the control over the variables of the thermomechanical process was lost.
c.  When evaluating the mechanical behavior during tension tests, it is concluded that the amount of retained austenite and its enrichment in carbon directly influences the elongation to fracture of the studied steels. The steel that exhibited the greatest fracture elongation was the M390-300, which presented a variation in the amount of retained austenite ($\gamma$) from 8.1% in vol. to 1.9% vol., with a fracture elongation of 23.2%, which evidenced the presence of the TRIP phenomenon in steel.
d.  From the anisotropy study carried out, it is concluded that as the temperature of the isothermal bainitic treatments increases, the gamma fiber (<111>//ND) decreases, which produces deterioration in the deep drawing behavior.
e.  From the thermal study of the stability of the retained austenite and its influence on mechanical properties, it is concluded that the M390-300 steel presented its maximum elongation of 30% at a temperature of 323 K (50 °C). This increase in elongation correlates with the TRIP effect present in steel, which is favored when working within the critical zone of temperatures ($M_s^\sigma$ and $M_d$).

**Author Contributions:** Conceptualization, A.G. and A.M.; methodology, A.G. and A.M.; software, A.G.; validation, A.G. and A.M.; formal analysis, A.G. and A.M.; investigation, A.G.; resources, A.M.; data curation, A.G.; writing—original draft preparation, A.G. and A.M.; writing—review and editing, A.G. and A.M.; visualization, A.G. and A.M.; supervision, A.M.; project administration, A.M.; funding acquisition, A.M. All authors have read and agreed to the published version of the manuscript.

**Funding:** This research was funded by FONDECYT, grant number 1170905 and the APC was funded by Engineering Faculty.

**Institutional Review Board Statement:** Not applicable.

**Informed Consent Statement:** Not applicable.

**Data Availability Statement:** The data presented in this study are available on request from the corresponding author.

**Acknowledgments:** Alberto Monsalve gratefully acknowledges to DICYT USACH. Alexis Guzmán gratefully acknowledges to VRIP UDA.

**Conflicts of Interest:** The authors declare no conflict of interest.

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
