# Peer review of "Effect of Bainitic Isothermal Treatment on the Microstructure and Mechanical Properties of a CMnSiAl TRIP Steel"

_metals, doi:10.3390/met12040655_

Round 1
Reviewer 1 Report
The paper is very well organized and written, which systematically investigates the effect of bainite transformation temperature on the microstructure evolution and mechanical properties, in terms of tensile properties, anisotropy test and high-temperature deformation. Additionally, the relationship between microstructure and mechanical properties is discussed in detail. This paper deserves publication in Metals and, however, several minor revisions are necessary. 1. The title does not represent the main content of the present study, which focuses on the design of heat treatment and the dependence of mechanical properties on the bainite holding temperature. But not mainly on the effect of substitution of Si by Al. 2. In the Abstract, “TRIP-assisted CMnSiAl fully martensitic steel”, it will make readers confused what fully martensitic means. Actually, martensite is the initial microstructure. 3. Line 81, “420 y 1000”, what does “y” mean? 4. Line 117-118, the description of chemical analysis should be put the place related to Table 1. 5. Figure 3b, the “Austenite” marked in the image should be “Ferrite”?? 6. Figure 11, the description of a,b,c is missed. 7. Line 323 in conclusion, the volume fraction of ferrite should be given.Author Response
Please find attached the responses to reviewer 1.

Reviewer 2 Report
- Comparing with other TRIP-assisted steel, the mechanical performance in this work is not well. This needs clear explanation.
- Some recent breakthorugh in this field must be reviewed, ex. https://doi.org/10.1016/j.msea.2019.06.015
- The reason for replacing Si by Al is not sufficient. The primary reason is about galvanizing.
- Elastic regime seems not normal in tensile test (Figure 12)
- Vertical axis should be total elongation in Figure 14(c), instead of strain.
- Figure resolution should be higher. Scale bar must be clear.
Author Response
Please find attached the answers to referee 2.

Reviewer 3 Report
The article highlights peculiarities of the influence of the heat treatment modes of steel and the effect of partial substitution of Si by Al on its microstructure and properties. However, insufficient attention is paid to the comparison of studied steel with standard TRIP 780 steel. It is not clear what the advantage of studied steel over standard TRIP 780 steel is, because a slightly higher value of its relative elongation (plasticity) corresponds to significantly lower values of strength characteristics (tensile strength and yield strength).
The article is interesting, but a number of shortcomings need to be corrected:
- In Fig.1, the font size should be increased.
- In Fig.9, the text cannot be recognized.
- In Fig.10 and Fig.11, the text cannot be recognized. Resolutions of the figures should be increased.
- There is an uncertainty about the linear part of a stress-strain diagram in Fig.12. For what reason does this part of the diagram vertical?
- The statement (Lines 275-276) “Due to M 390-300 sample showed the best mechanical results, a thermal study of the stability of the retained austenite was carried out in order to understand its influence on the mechanical properties.” is controversial, since this material showed the highest relative elongation, but is significantly inferior to other steels in tensile strength and yield strength.
- Please clearly substantiate the advantage of experimental steel over standard TRIP 780 steel.
Author Response
Please find attached the file with the answers to reviewer 3.

Reviewer 4 Report
The manuscript is original and interesting, well written with sound conclusions.
It deserves to be published in the present form after a minor correction: in the experimental section the authors should define how XRD experiments have been carried out (X-ray wavelength, step, counting time per step etc.).
Author Response
Please find attached the answers to referee 4

Reviewer 5 Report
The article is devoted to the study of TRIP steels. A lot of experimental work has been done. Modern methods and equipment for the study were used.
However, when reviewing the article, I had questions and comments.
- The title of the article does not reflect the content of the work. Based on the name, we should talk about the effect of Al on the microstructure and properties of TRIP steel, and in the article much attention is paid to the heat treatment regime and there are no studies devoted to assessing the effect of Al. Perhaps the authors inaccurately focused the attention of readers in their reasoning. The title of the article and the conclusion must be agreed upon. Now there are many descriptions of heat treatment modes in the conclusions.
- The introduction to the article is very short and does not reflect all the research that has been done on this topic before. A total of 7 references to previous studies. It is necessary to conduct a literature review and indicate how your research differs from previously performed ones.
- What is the rationale for choosing the content of Al=0.43%? Why not more or less?
- Table 1 shows the chemical composition of the studied steels. How was it defined? What equipment and methods were used? This information occurs much later (lines 117-118). You need to indicate this next to table 1.
- For clarity and a better understanding of the technology for producing cold-rolled sheets, I recommend to give a drawing with a sequence of operations and an indication of the modes.
- In Figure 1, the phase indicators and their names need to be increased. Now they are small and unreadable. For clarity, add temperatures (950, 750, 420) to the heat treatment graph.
- Specify a setup name for differential scanning calorimetry (DSC) you used.
- The article does not reflect the results of a study of the influence of the content of Si and Al on the microstructure and properties of steel.
Author Response
Please find enclosed the answers to referee 5

Round 2
Reviewer 3 Report
The authors took into account the comments of the reviewer and made appropriate corrections to the manuscript. The article is interesting and can be accepted in present form.
Reviewer 5 Report
The authors made the additions recommended by me to the article and corrected the existing shortcomings. I recommend the article in the presented version for publication in the journal.